# Association of Vaccine Confidence and Hesitancy in Three Phases of COVID-19 Vaccine Approval and Introduction in Japan

**DOI:** 10.3390/vaccines10030423

**Published:** 2022-03-10

**Authors:** Mikiko Tokiya, Megumi Hara, Akiko Matsumoto, Mohammad Said Ashenagar, Takashi Nakano, Yoshio Hirota

**Affiliations:** 1Department of Social and Environmental Medicine, Faculty of Medicine, Saga University, 5-1-1 Nabeshima, Saga 849-8501, Japan; matsumoa@gmail.com (A.M.); sx5080@cc.saga-u.ac.jp (M.S.A.); 2Department of Preventive Medicine, Faculty of Medicine, Saga University, 5-1-1 Nabeshima, Saga 849-8501, Japan; harameg@cc.saga-u.ac.jp; 3Department of Pediatrics, Kawasaki Medical School, 577 Matsushima, Kurashiki 701-0192, Japan; ndhkk029@ybb.ne.jp; 4Clinical Epidemiology Research Center, Medical Co. LTA (SOUSEIKAI), Higashi-ku, Fukuoka 813-0017, Japan; hiro8yoshi@lta-med.com

**Keywords:** vaccine, hesitancy, vaccine confidence, vaccine literacy, risk communication, phases of COVID-19 vaccine introduction

## Abstract

Understanding vaccine hesitancy, considering the target region and phase, is an urgent issue to quell the coronavirus disease (COVID-19) pandemic. This study aimed to monitor COVID-19 vaccine hesitancy in the Japanese population during the three phases of vaccine approval and introduction, and evaluate the association of vaccine hesitancy with vaccine confidence and literacy. We conducted web-based cross-sectional surveys during the three phases of COVID-19 vaccine introduction: January 2021, before approval; June, start of vaccination of the elderly; and September, when about 70% of the target population was vaccinated with at least one dose. There were 7210 participants, aged 20–80 years. We evaluated the association of vaccine hesitancy with vaccine confidence and literacy in the three phases using multivariate logistic regression analysis. The proportion of hesitancy in January, June, and September was 17.5%, 65.3%, and 19.4%, respectively. In any phase, lower vaccine confidence and literacy showed a higher adjusted odds ratio (AOR) of vaccine hesitancy in most items (AOR > 1, *p* < 0.001). Vaccine hesitancy in June had a different trend in perception of COVID-19 compared to that in the January and September surveys. The findings suggested that hesitancy increases transiently during vaccination introduction phases, and changes as the vaccination program progressed or waves of epidemic. Careful risk communication to increase vaccine confidence and literacy is essential to reduce vaccine hesitancy, especially in the introduction phase.

## 1. Introduction

Focusing on vaccine hesitancy is helpful for risk communicators and policy makers to promote vaccination [1]. Vaccine hesitancy is defined as a delay or refusal to accept vaccination despite the availability of immunization services [2] and is described by the WHO as a threat to health [3]. Public health risk communication is the systematic dissemination of information to diverse audiences about the existence, nature, and severity of risks and hazards that affect their health, safety, and environment so that they can make informed and proactive decisions [4]. Understanding a subject’s situation and applying it to communication is an urgent issue.

COVID-19 vaccines have been put into emergency use worldwide to control the pandemic. According to a survey conducted about six months prior to the start of vaccination, vaccine hesitancy in the United States and Japan was 14% (May 2020, *n* = 2006, probably not willing and not willing) [5] and 12%, respectively [6]. According to previous studies [7], intentions of COVID-19 vaccination have been reported to be influenced by time since the start of the pandemic, sex, age, educational and socioeconomic factors, region, religion, trust in the government, trust in the vaccine, and recommendations by health care providers. In addition, it has been reported that the proportion of vaccine hesitancy has increased with the news of adverse events and safety concerns [7]. In addition, vaccine hesitancy varies by region, time, and vaccine type [2]. Therefore, to end the current pandemic and prepare for the future epidemics, it is necessary to understand the situation of vaccine hesitancy, considering a targeted region and time and use this information in communications.

In Japan, several waves of the epidemic have been observed since the emergence of the first cases, confirmed in January 2020. In parallel, free vaccines have been prepared by the government, and vaccination for health care providers started in February 2021 [8]. Target populations included the elderly, those with underlying medical conditions, and the general population older than 12 years old, in that order. Information on COVID-19 varies from official websites and public news to personal dissemination, and the situation surrounding COVID-19 is ever-changing. However, no studies have been conducted to investigate the association of vaccine confidence and literacy with vaccine hesitancy at each wave of the epidemic.

We hypothesized that different phases of vaccine introduction or epidemic waves would have different degrees of hesitancy, and that vaccine confidence and literacy would differ between those with and without hesitancy. The purpose of this study was to clarify the above hypothesis, and to use the study results for risk communication.

## 2. Materials and Methods

### 2.1. Survey Sample and Data Collection

We conducted web-based cross-sectional surveys during the three phases of COVID-19 vaccine introduction. The first survey was conducted on 19–20 January 2021, before approval [9]. The second survey was conducted on 23–24 June, when vaccination started for the elderly and when 19.2% of the target population in Japan had received at least one dose of vaccine (June survey) [10]. The third survey was conducted 27–29 September, when 66.5% of respondents had received at least one dose of vaccine (September) [10]. These three periods were just after the third, fourth, and fifth waves of the COVID-19 epidemic in Japan (Appendix A). The details of sample and data collection were included in the report on the first survey [9]. Approximately 1.2 million survey participants were registered in the panel of a web-survey company (Macromill, Inc., Tokyo, Japan). Participation in the panel was voluntary, and points were given to those who participated in the survey. These points could be used to buy products and services from corporate sponsors. The survey targets were 7000 men and women, aged 20–80. The survey participants were registered via email and an app until the target number of participants was reached. Instructions for the survey were sent via email. The questionnaire was uploaded in a secure section of the website and participants were asked to answer each question to ensure that no variables were missing. In addition, participants were informed: “The content of your answers may be made public through statistical processing in a form that does not identify you personally. Thank you for your understanding”. Answering the questionnaire was regarded as evidence of consent. This study was approved by the ethics committee of Saga University, Saga, Japan (No: R2-24).

### 2.2. Methods of Measurement

#### 2.2.1. Definition of COVID-19 Vaccine Hesitancy

In this study, COVID-19 vaccine hesitancy was defined as follows: in the January survey, “If the COVID-19 vaccine were approved, would you want to be vaccinated?” to which respondents answered on a five-point Likert scale of “strongly disagree, disagree, neither agree nor disagree, agree, and strongly agree”. We defined “strongly disagree” and “disagree” as “hesitancy”. In the June and September surveys, “hesitancy” was defined as those who had not received at least one dose of vaccination at the time of the survey and did not plan to do so in the future (those who did not fall into the categories of “have not received any information” or “have no appointment date”); it was not because they did not have vaccine available for their age group (Appendix A).

#### 2.2.2. Assessment of Sociodemographic Factors

Sociodemographic factors included sex, age group (20–29, 30–39, 40–49, 50–59, 60–69, and 70–80 years), occupation, region of residence, marital status (married or unmarried), number of children, household income category (less than 2 million yen, 2–4 million yen, and more than 4 million yen), and educational attainment.

#### 2.2.3. Assessing Vaccine Confidence and Literacy

In this study, 14 questions were asked based on validated scales and indicators to measure confidence in vaccines and literacy [7,9,11,12]. The following topics were included: importance, effectiveness, safety, reduced chance of infection, serious adverse reactions, dangers of new vaccines, concern about serious adverse reactions, hassle, no need to vaccinate against infections that have become rare, voluntary vaccines do not need to be given, if others around you get vaccinated, you don’t have to get vaccinated, vaccine information is easy to obtain, understanding of vaccinations received in the past.

The responses were categorized on five levels (“Not at all agree” to “Fairly agree”). To measure the level of trust and literacy in vaccines, the following scores were given: strongly disagree = 1 point, disagree = 2 points, undecided = 3 points, agree = 4 points, and strongly agree = 5 points.

#### 2.2.4. Other Relevant Factors

**Recognition of COVID-19 (8 items).** The following eight questions were asked to determine what the participants recognized about the COVID-19. The answers were obtained on a five-point Likert-type rating scale ranging from “strongly disagree” to “strongly agree”, as described. (1) “I know a lot about COVID-19”, (2) “All COVID-19 patients have symptoms”, (3) “Many people with COVID-19 have mild symptoms (mild illnesses)”, (4) “ COVID-19 are more severe in people over 65 years old and those with chronic illnesses”, (5) “COVID-19 is easily spread from person to person”, (easily infectious), (6) “I am worried about getting a COVID-19 (worried about getting)”, (7) “ I may get a COVID-19 (may get)”, and (8) “Once you have a COVID-19, you cannot get it again”.

**Awareness of the COVID-19 vaccine (6 items).** The following six questions were asked to find out what the participants thought about COVID-19 vaccines. Responses were obtained on a 5-point Likert-type rating scale ranging from “strongly disagree” to “strongly agree”, as described. (1) “ Preventing inoculated individuals from becoming seriously ill with COVID-19”, (2) “ Prevention of COVID-19 in vaccinated persons”, (3) “Prevent family members and friends of the inoculated person from contracting COVID-19”, (4) “Prevent the spread of COVID-19 in the vaccinated person’s area”, (5) “ I am concerned about adverse reactions to the COVID-19 vaccine”, and (6) “You may experience fever or swelling at the vaccination site after receiving the COVID-19 vaccine”.

### 2.3. Statistical Analysis

Cross-sectional analyses were conducted for the three surveys in January, June, and September. All tests were conducted using categorical variables. Each month, we assessed factors associated with vaccine hesitancy using χ^2^ tests. Multivariate logistic regression analysis adjusted for age and sex was conducted for each survey month to explore the relationship between vaccine confidence and literacy. The dependent variable was the presence or absence of vaccine hesitancy. The explanatory variables were vaccine confidence and literacy (14 items), awareness of COVID-19 (8 items), and awareness of COVID-19 vaccines (6 items). Statistical significance was set at two-sided *p* < 0.05 for the χ^2^ test. The Bonferroni method was used in multivariate logistic regression analysis, and the significance level was set at *p* < 0.001. The sample size was calculated to be between 2000 and 5000 (alpha = 0.05/number of survey items (50 items) = 0.001; beta = 0.20; odds ratio, 1.5; vaccination hesitancy, 30–50%; and percentage of possession of relevant factors, 10–20%). SAS version 9.4 (SAS Institute Inc., Cary, NC, USA) was used for statistical analyses.

## 3. Results

Table 1 shows the characteristics of the respondents for each survey. The total number of participants each month was 7210. There were no significant differences in participants’ sex, age, region, marital status, number of children, household income, educational attainment, body mass index, smoking, underlying disease, and healthcare worker status. The proportion of vaccine uptake increased after COVID-19 vaccine was approved and introduced to Japanese population. About 18% and 70% of the participants were vaccinated with two doses in the June and September surveys, respectively. The proportion of subjects among unvaccinated who answered, “strongly disagree” and “disagree” to the question “If the COVID-19 vaccine was available, would you want to be vaccinated?” increased as the survey period progressed.

Table 2 shows the proportion of vaccine hesitancy according to participant characteristics in each survey period. A total of 1264 participants (17.5%) were hesitant in January, 4708 (65.3%) in June, and 1396 (19.4%) in the September. In the comparison between January and June (Jan:June), the proportion of hesitancy in each attribute was significantly higher in June (*p* < 0.001), except for health care providers. The proportion of hesitancy among health care workers did not change between January and June (Jan 17.0%, June 18.9%, *p* = 0.734). There was no significant difference in the percentage of hesitancy according to demographics (*p* > 0.05), except for age (20s) and health care providers. Hesitancy in participants among 20s was significantly more common in the September survey (34.0%, *p* = 0.025) than in the January survey (19.1%). There were significantly fewer healthcare providers in the September survey (5.4%, *p* = 0.009) than in the January survey (17.1%).

Table 3 shows the results of multivariate logistic regression analysis for the evaluation of vaccine confidence and literacy, awareness of COVID-19, and awareness of COVID-19 vaccines, with the presence or absence of vaccine hesitancy as the dependent variable for each survey period. In the category of vaccine confidence and literacy, the same trend was observed in all three surveys. Hesitancy respondents had significantly lower odds ratios (*p* < 0.001) of agreeing with the following seven items: importance of vaccination, efficacy, safety, availability of information on vaccination, necessity, and understanding of past vaccinations. The odds ratio of “agree” was significantly higher (*p* < 0.001) among hesitancy respondents for the following seven items: having an adverse reaction to vaccination, risk of new vaccines, worrying about adverse reactions, cumbersome vaccination, not needing vaccines for rare diseases, not needing voluntary vaccines, and not needing vaccines if others around them were vaccinated.

In the category of recognition of COVID-19, there was no significant difference between vaccine hesitancy and acceptance in the June survey in terms of mild illnesses, worried about getting, and may get. Additionally, June vaccine hesitancy had a significantly higher odds ratio of “Agree” in the statement “Once you have COVID-19 infection, you cannot get it again”.

In the category of recognition of COVID-19 vaccine, fever and swelling after vaccination was different in three surveys. Vaccine hesitancy in the June survey had a significantly lower odds ratio of “Agree” in “fever and swelling after vaccination”. Hesitancy had significantly lower odds ratios in the following items: “prevention from becoming seriously ill”, “prevention of infection in vaccinated persons”, and “prevent the spread in the vaccinated person’s area” (*p* < 0.001). In contrast, hesitancy had a significantly higher odds ratios in “concern about adverse reactions” (*p* < 0.001).

## 4. Discussion

This study investigated the COVID-19 vaccine hesitancy in Japanese population during the three phases of vaccine approval and introduction, and evaluated the association of vaccine hesitancy with vaccine confidence and literacy in the three phases. The results of the study showed that hesitancy increased transiently during epidemics and vaccination introduction phases; however, the association between hesitancy, vaccine confidence and literacy were similar. In any survey, lower vaccine confidence and literacy showed a higher AOR of vaccine hesitancy. Careful risk communication, including the disclosure of negative information, is needed.

As expected, proportion of hesitancy was different in the phases of the vaccine introduction or epidemic waves. However, vaccine confidence and literacy had a similar trend in each survey. That is, a higher percentage of vaccine hesitancy was found in June, when vaccination among the elderly was started, compared to January, before approval; and September, when about 70% of target population were vaccinated with at least one dose (17.5% (January), 65.3% (June), 19.4% (September)). Hesitancy in June also showed different trends in the “Recognition of COVID-19” and “Recognition of COVID-19 vaccines” compared to the January and September surveys. In a cohort study of 4654 adults in the U.S., pre-vaccine approval hesitancy was about 31%, but at the end of the vaccination period, about 70% of the hesitancy adults were vaccinated with one or more doses or intended to get vaccinated [13]. Siegler et al. [13] stated that vaccine hesitancy changes over time for unknown reasons. In our study, two-thirds of the subjects had vaccine hesitancy in the June survey. We believe that this supports the change in hesitancy over time. In addition, during the epidemic and vaccination progress, adverse events have been also reported but cannot be evaluated in relation to the vaccine [11,14]. In a survey conducted in Australia before the start of vaccination, the reasons for vaccine hesitancy were safety concerns followed by lack of decision-making power [15]. These previous studies suggest that even in cases of vaccine hesitancy, if there is information that allows the decision to be made, there is a possibility that the vaccine hesitancy will change to vaccination. It is important to know why they are hesitating and address the concerns.

In the three phases of COVID-19 vaccine introduction, vaccine hesitancy among younger generations (20s) showed a different trend from the other generations (Appendix A). In other words, while there was no significant difference in the percentage of hesitancy between the January and September surveys, one-third of those in their 20s were hesitant, even in the September survey. This trend of being more hesitant in 20s is consistent with previous studies [16]. There are four possible reasons for this. One is the concern about adverse effects. This may be due to many reports of adverse events in young people [17], such as cases of vaccine-related myocarditis and pericarditis reported in Japan and abroad [10,18,19]. Second, negative information due to adverse reactions may also have an impact. Huangfu et al. [20] showed an association between the magnitude of concern about adverse reactions to vaccines and hesitancy from an analysis in Twitter [20]. In a study that analyzed Twitter about COVID-19, the highest number of tweets were negative [21]. In addition, several previous studies have reported that negative information gives people a stronger impression than positive information [22]. Third, there may be concerns about the effects on reproduction and the next generation, as a systematic review of the influenza vaccine hesitancy in pregnant women reported concerns about safety and risks to mother and child [23]. Therefore, we believe that long-term reproductive and next-generation concerns are not something that can be resolved in a short period. To address vaccine hesitancy in the younger generation, as Bozzola et al., mentioned [24], it is important to balance facts with positive messages by using multiple communication channels, such as monitoring information needs online, using digital communication through TikTok and Instagram, which are apps used by young people, and engaging interdisciplinary groups, including the general public, in communication channels [24]. The fourth reason is that as the time for vaccination approaches, people may become temporarily hesitant to get vaccinated. In Japan, citizens between the ages of 12 and 64 were notified by the end of July that they should be vaccinated by the end of September [8]. The notifications were provided and vaccination was conducted in order of age, starting with the oldest group [8]. Therefore, it is possible that at the timing of the September survey, may having been shortly before some participants were scheduled for their vaccination. According to data released afterwards, 76.4% of people in their 20s had received at least one dose of vaccination as of December (Appendix A). In the future, it will be necessary to conduct a similar survey when vaccination of people in their 20s has progressed to confirm vaccine hesitancy.

In this survey, the low rate of vaccine hesitancy among health care providers was similar to previous studies [25]. However, the rates in the three phases showed a different trend from the overall trend. Hesitancy in the June survey, when the overall hesitancy was about 60%, was the same as the hesitancy in the January survey. Furthermore, although there was no significant difference in the overall of hesitancy between the January and the September. Nonetheless, the percentage of hesitancy was significantly lower among healthcare providers in the September than in the January. In addition, unlike the overall results, there was no significant difference in hesitancy among medical personnel in the items “serious adverse reactions”, “troublesome”, and “unnecessary” in this survey (Appendix A). The vaccination status of health care providers is said to affect the public [26,27]. The role of healthcare providers in vaccine deployment has also been reported to be beneficial [13]. In our previous study, physician recommendation was cited as a factor influencing vaccination in the general population [9]. Similarly, trust in health care providers has been reported to be a strong predictor of COVID-19 vaccine acceptance [28], and vaccine hesitancy has been reported to increase as a result of mistrust in health care [29]. In Japan, approximately 80% of the population has been vaccinated by December 2021, despite it being non-mandatory and its late start [30]. We believe that this may be because the number of vaccine hesitancy in the health care provider was small as of the September.

The definition of vaccine hesitancy in the June and September surveys differs from the definition in the January survey. However, we believe that the impact of this difference is small. The SAGE Group defines vaccine hesitancy as delaying or refusing to be immunized despite the availability of immunization services [2]. This time, in the January and June surveys (or the September survey), the questions were asked differently, but both asked what the SAGE group defines as hesitancy. We believe that we have a good grasp of hesitancy.

This study showed that the perceptions of vaccine acceptance and vaccine hesitancy conflicted. The items “I know a lot about COVID-19 infections”, “Most people with COVID-19 infections are 65 years old or older, or have a chronic illness”, and “Infectious diseases spread easily from person-to-person”, were negative in all three surveys. In contrast, the questions related to “adverse reactions”, “risks”, “hassle”, and “no need for vaccine” had positive responses. These perceptions are at odds with the official government communication that recommends vaccines. Low trust in government has been reported to be associated with COVID-19 vaccine rejection [31]. Public health risk communication messages must convey accurate and objective knowledge and understanding of risks and hazards that affect health to diverse audiences [4]. Information about diseases should include clear evidence of transmission, infectiousness, and the number of people who have died from the disease. Information about the vaccine should include the effectiveness of the vaccine in preventing infection, serious illness, and death, as well as possible adverse effects [32] and their probability of occurring if the vaccine is given. Moreover, there must be an expectation that the government will take responsibility for addressing adverse reactions that occur after vaccination [33]. It has been reported that failures in risk communication result from ‘information vacuums’ and can be filled with biased sources of information or sources that may not accurately communicate the risks [4]. For these reasons, regulators need to listen to the public [34], be transparent in their assessment of data on vaccine quality, safety, and efficacy, and have a responsibility to maintain continuous and close safety oversight and advice to the public on safe use [35]. Moreover, the SAGE Working Group stated that civil society, community organizations, and the private sector should demand and monitor specific actions from manufacturers, governments, and other stakeholders [36]. In Japan, it is difficult to say that there is sufficient communication about vaccines [37]. There is a need to tailor the message to the target audience to increase the confidence in vaccines and help the public in making appropriate decisions [38,39]. In the future, in cases of necessary vaccinations, communication should consider the anxiety and other emotions that hesitancy people may have [40,41].

The study has several limitations. First, it was a web-based survey; the people surveyed had easy access to the Internet and may have been exposed to much information through the Internet. Therefore, it cannot be denied that there was selection bias and sampling bias. However, those who completed the questionnaire were given points that could be used to purchase products and services from partner companies, so not only those interested in the COVID-19 vaccine participated. Second, the subjects of the three surveys are not the same, so it was not possible to examine changes over time. Third, it has been reported that temperature or air quality have a significant impact on COVID-19 mortality [42]; however, the influence of temperature and/or air quality on vaccine hesitancy and COVID-19 perception was not examined in the present study. Although regional differences in temperature exist in Japan, the temperature in June and September is similar within each region [43]. Thus, influence of temperature on present results may be minimal. Despite these limitations, to the best of our knowledge, this is the first study to capture the confidence and literacy of vaccine hesitancy in the three phases of vaccine approval and introduction.

## 5. Conclusions

This study suggested that hesitancy increases transiently during vaccination introduction phases, and changes as the vaccination program progressed or waves of epidemic. In any phase, lower vaccine confidence and literacy showed a higher AOR of vaccine hesitancy. Careful risk communication to increase vaccine confidence and literacy is essential to reduce vaccine hesitancy, especially in the introduction phase.

## Figures and Tables

**Table 1 vaccines-10-00423-t001:** Characteristics of participants.

		January (*n* = 7210)	June (*n* = 7210)	September (*n* = 7210)	
		*n* (%)	*n* (%)	*n* (%)	*p*-Value
Sex	Female	3815 (52.9)	3827(53.1)	3823 (53.0)	0.979
Age	20–29	1110 (15.4)	1073 (14.9)	1036 (14.4)	0.935
	30–39	1312 (18.2)	1309 (18.2)	1323 (18.4)	
	40–49	1356 (18.8)	1345 (18.7)	1334 (18.5)	
	50–59	1274 (17.7)	1303 (18.1)	1321 (18.3)	
	60–69	1116 (15.5)	1113 (15.4)	1121 (15.6)	
	70–80	1042 (14.5)	1067 (14.8)	1075 (14.9)	
Area	Hokkaido	311 (4.3)	345 (4.8)	341 (4.7)	0.992
	Tohoku	371 (5.2)	393 (5.5)	385 (5.3)	
	Kanto	2788 (38.7)	2737 (38.0)	2757 (38.2)	
	Chubu	1199 (16.3)	1181 (16.4)	1162 (16.1)	
	Kinki	1410 (19.6)	1413 (19.6)	1415 (19.6)	
	Chugoku	364 (5.1)	354 (4.9)	356 (4.9)	
	Shikoku	179 (2.5)	180 (2.5)	182 (2.5)	
	Kyusyu	588 (8.2)	607 (8.4)	612 (8.5)	
Married	Yes	4501 (62.4)	4441 (61.6)	4427 (61.4)	0.403
Child	Yes	4186 (58.1)	4132 (57.3)	4133 (57.3)	0.582
Household income	<4 million yen	1876 (26.2)	1927 (26.7)	1945 (27.0)	0.574
	≥4 million yen	3632 (50.4)	3644 (50.5)	3622 (50.2)	
	unknown	1702 (23.6)	1639 (22.7)	1643 (22.8)	
Educational attainment	High School Graduate	2118 (29.4)	2111 (29.3)	2108 (29.2)	0.983
	Above Higher Education	5092 (70.6)	5099 (70.7)	5102 (70.8)	
Obesity	Yes	1411 (19.6)	1314 (18.2)	1308 (18.1)	0.050
Smoking	Yes	1151 (16.0)	1137 (15.8)	1113 (15.4)	0.679
Underlying disease	Yes	2573 (35.7)	2450 (34.0)	2488 (34.5)	0.083
Health care worker	Yes	808 (11.2)	824 (11.4)	830 (11.5)	0.837
COVID-19 vaccine	Two time	0 (0)	1288 (17.9)	5009 (69.5)	<0.001
	One time	0 (0)	858 (11.9)	799 (11.1)	
	Unvaccinated	7210 (100)	5064 (70.2)	1402 (19.5)	
vaccine hesitancy * in Unvaccinated		1264 (17.5)	4708 (93.0)	1396 (99.6)	<0.001

* Subjects among unvaccinated who answered, “strongly disagree” and “disagree” to the question “If the COVID-19 vaccine were available, would you want to be vaccinated?”.

**Table 2 vaccines-10-00423-t002:** Difference in hesitancy according to the participants characteristics by survey month.

		January (*n* = 7210)	June (*n* = 7210)	September (*n* = 7210)	All	Jan. vs. Jun	Jan. vs. Sep
		*n* (%)	*n* (%)	*n* (%)	*p*-Value		
Hesitancy	Yes	1264 (17.5)	4708 (65.3)	1396 (19.4)	<0.001	<0.001	0.739
Sex	Male	527 (15.5)	2320 (65.6)	655 (19.3)	<0.001	<0.001	0.477
	Female	737 (19.3)	2388 (62.4)	741 (19.4)	<0.001	<0.001	0.991
Age	20–29	221 (19.1)	851 (79.3)	352 (34.0)	<0.001	<0.001	0.025
	30–39	268 (20.4)	924 (70.6)	368 (27.8)	<0.001	<0.001	0.222
	40–49	269 (19.8)	931 (69.2)	280 (21.0)	<0.001	<0.001	0.840
	50–59	224 (17.6)	962 (73.8)	213 (16.1)	<0.001	<0.001	0.734
	60–69	154 (13.8)	672 (60.4)	110 (9.8)	<0.001	<0.001	0.382
	70–80	128 (12.3)	368 (34.5)	73 (6.8)	<0.001	<0.001	0.186
Area	Hokkaido	52 (16.7)	217 (62.9)	74 (21.7)	0.009	<0.001	0.371
	Tohoku	46 (12.4)	258 (65.7)	75 (19.5)	<0.001	<0.001	0.192
	Kanto	476 (17.1)	1853 (67.7)	507 (18.4)	<0.001	<0.001	0.807
	Chubu	218 (18.2)	778 (65.9)	256 (22.0)	<0.001	<0.001	0.497
	Kinki	274 (19.4)	890 (63.0)	250 (17.7)	<0.001	<0.001	0.749
	Chugoku	71 (19.5)	216 (61.0)	66 (18.5)	<0.001	<0.001	0.861
	Shikoku	31 (17.3)	116 (64.4)	37 (20.3)	<0.001	<0.001	0.586
	Kyusyu	96 (16.3)	380 (62.6)	131 (21.4)	<0.001	<0.001	0.359
Married	No	533 (19.7)	1947 (70.3)	749 (26.9)	<0.001	<0.001	0.227
	Yes	731 (16.2)	2761 (62.2)	647 (14.6)	<0.001	<0.001	0.750
Child	No	617 (20.4)	2252 (73.2)	840 (27.3)	<0.001	<0.001	0.252
	Yes	647 (15.5)	2456 (59.4)	556 (13.5)	<0.001	<0.001	0.686
Household income	<4 million yen	330 (17.6)	1261 (65.4)	444 (22.8)	<0.001	<0.001	0.356
	≥4 million yen	605 (16.7)	2326 (63.8)	577(15.9)	<0.001	<0.001	0.889
	unknown	329 (19.3)	1121 (68.4)	375 (22.8)	<0.001	<0.001	0.545
Educational attainment	High School Graduate	353 (16.7)	1486 (70.4)	516(24.5)	<0.001	<0.001	0.172
	Above Higher Education	911 (17.9)	3222 (63.2)	880 (17.3)	<0.001	<0.001	0.905
Obesity	No	1051 (18.1)	3809 (64.6)	1168 (19.8)	<0.001	<0.001	0.763
	Yes	213 (15.1)	899 (68.4)	228 (17.4)	<0.001	<0.001	0.655
Smoking	No	1102 (18.2)	3872 (63.8)	1126 (18.5)	<0.001	<0.001	0.959
	Yes	162 (14.1)	836 (73.5)	270 (24.3)	<0.001	<0.001	0.067
Underlying disease	No	868 (18.7)	3304 (69.4)	1030 (21.8)	<0.001	<0.001	0.587
	Yes	396 (15.4)	1404 (57.3)	366 (14.7)	<0.001	<0.001	0.893
Heath care worker	No	1126 (17.6)	4552 (71.3)	1351 (21.2)	<0.001	<0.001	0.521
	Yes	138 (17.1)	156 (18.9)	45 (5.4)	0.011	0.734	0.009

In January, “hesitancy” was defined as a response of “strongly disagree” or “disagree” to the question “If a COVID-19 vaccine becomes available, I will get vaccinated”. In June and September, “hesitancy” was defined as those who had not yet vaccinated and no plans to get the vaccine at the time of the survey (Appendix A). “Obesity” was defined as BMI ≥ 25. Health care providers includes physicians, dentists, veterinarians, pharmacists, public health nurses, nutritionists, nurses, nurse assistants, physical therapists, occupational therapists, and dental technicians. Medical office and nursing care workers were not included in this group. The *p*-values were calculated by the χ^2^ test. All three months (January, June, and September): Data were calculated based on the presence or absence of “hesitancy” using the 2 × 6 χ^2^ test. January:June: Data were calculated based on the presence or absence of “hesitancy” in January and June using the 2 × 2 χ^2^ test. January:September: Data were calculated based on the presence or absence of “hesitancy” in January and September using the 2 × 2 χ^2^ test.

**Table 3 vaccines-10-00423-t003:** Age and sex adjusted odds ratio (OR) of vaccine hesitancy according to vaccine confidence and literacy by phase.

	January			June			September		
	AOR	95% CI	*p* Value	AOR	95% CI	*p* Value	AOR	95% CI	*p* Value
**Vaccine confidence and literacy**									
Vaccines are important for my health	0.31	0.28–0.33	<0.001	0.57	0.53–0.61	<0.001	0.34	0.31–0.37	<0.001
Vaccines are effective	0.37	0.34–0.41	<0.001	0.59	0.55–0.64	<0.001	0.36	0.33–0.39	<0.001
Vaccines are safe	0.36	0.33–0.39	<0.001	0.73	0.69–0.77	<0.001	0.41	0.38–0.44	<0.001
My vaccination is important for the health of others in my community	0.69	0.55–0.63	<0.001	0.71	0.69–0.76	<0.001	0.50	0.46–0.53	<0.001
I am concerned about serious adverse effects of vaccines	1.87	1.72–2.03	<0.001	1.14	1.05–1.18	<0.001	1.56	1.44–1.68	<0.001
New vaccines carry more risks than older vaccines	1.87	1.73–2.02	<0.001	1.24	1.17–1.32	<0.001	1.85	1.72–2.00	<0.001
Serious adverse reactions may occur due to the vaccination	1.82	1.69–1.97	<0.001	1.56	1.48–1.64	<0.001	2.28	2.11–2.45	<0.001
I have difficulty getting immunized (no time, far medical institutions, etc.)	1.48	1.40–1.57	<0.001	1.56	1.48–1.65	<0.001	1.86	1.76–1.97	<0.001
I do not need vaccines for diseases that are not common anymore	1.30	1.21–1.39	<0.001	1.35	1.28–1.43	<0.001	1.55	1.45–1.65	<0.001
It is not necessary to take voluntary vaccination	2.24	2.08–2.41	<0.001	1.42	1.35–1.50	<0.001	2.26	2.11–2.42	<0.001
I do not need vaccines if everyone around me is immunized	1.83	1.71–1.95	<0.001	1.63	1.53–1.73	<0.001	2.21	2.07–2.36	<0.001
It is easy to obtain correct information on immunization	0.66	0.62–0.71	<0.001	0.75	0.71–0.80	<0.001	0.60	0.56–0.64	<0.001
It is easy to understand why immunization is needed.	0.63	0.59–0.67	<0.001	0.75	0.71–0.80	<0.001	0.62	0.58–0.66	<0.001
I have been able to accurately understand the vaccinations I have received	0.79	0.74–0.84	<0.001	0.59	0.55–0.62	<0.001	0.56	0.53–0.60	<0.001
**Recognition of COVID-19**									
I know a lot about COVID-19	0.88	0.82–0.95	<0.001	0.73	0.69–0.77	<0.001	0.77	0.72–0.83	<0.001
Anyone with COVID-19 will have symptoms.	0.86	0.81–0.93	<0.001	1.07	1.01–1.13	0.003	1.10	1.03–1.17	0.006
Many people with COVID-19 have mild illnesses.	1.09	1.03–1.16	0.003	1.04	0.99–1.09	0.139	1.16	1.10–1.23	<0.001
COVID-19 is more severe in people over 65 years old and those with chronic illnesses.	0.80	0.75–0.87	<0.001	0.82	0.77–0.88	<0.001	0.81	0.7–0.87	<0.001
COVID-19 easily spreads from person to person.	0.79	0.73–0.85	<0.001	0.89	0.83–0.94	<0.001	0.83	0.77–0.89	<0.001
I am worried about getting COVID-19.	0.66	0.62–0.71	<0.001	0.99	0.93–1.04	0.602	0.74	0.70–0.80	<0.001
I may get a new type of COVID-19.	0.80	0.75–0.86	<0.001	1.01	0.95–1.07	0.743	0.75	0.70–0.81	<0.001
Once you have COVID-19, you cannot get it again.	0.87	0.80–0.94	<0.001	1.10	1.03–1.18	0.004	1.03	0.95–1.12	0.513
**Recognition of COVID-19 vaccine**									
Preventing inoculated individuals from becoming seriously ill with COVID-19.	0.47	0.44–0.51	<0.001	0.70	0.65–0.76	<0.001	0.57	0.53–0.61	<0.001
Prevention of COVID-19 in vaccinated persons.	0.49	0.46–0.52	<0.001	0.84	0.79–0.89	<0.001	0.65	0.61–0.69	<0.001
Prevent family members and friends of the inoculated person from contracting COVID-19.	0.58	0.54–0.61	<0.001	0.85	0.81–0.90	<0.001	0.65	0.61–0.69	<0.001
Prevent the spread of COVID-19 in the vaccinated person’s area.	0.45	0.42–0.48	<0.001	0.73	0.69–0.78	<0.001	0.56	0.53–0.60	<0.001
I am concerned about adverse reactions to the COVID-19 vaccine.	2.09	1.92–2.30	<0.001	1.57	1.49–1.65	<0.001	2.67	2.46–2.90	<0.001
You may experience fever or swelling at the vaccination site after receiving the COVID-19 vaccine.	1.32	1.20–1.43	<0.001	0.84	0.78–0.90	<0.001	1.02	0.94–1.10	0.634

AOR: Adjusted for age and sex. The significance level was set at 0.05/14 = 0.0036 for the category of vaccine confidence and literacy and 0.05/8 = 0.0063 for the category of Recognition of COVID-19 using the Bonferroni method. For the category of Recognition of COVID-19 vaccine, the value was 0.05/6 = 0.0084.

## Data Availability

The data presented in this study are available on request from the corresponding author (M.T.). The data are not publicly available due to privacy concerns.

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
