# Peer review of "Association of Vaccine Confidence and Hesitancy in Three Phases of COVID-19 Vaccine Approval and Introduction in Japan"

_vaccines, 2022, doi:10.3390/vaccines10030423_

Round 1

Reviewer 1 Report

The manuscript byTokiya and colleagues investigates the vaccine hesitancy among a subset of the Japanese population (7,210 individuals) from ages 20-80. The investigators measured vccine hesitancy and literacy at three different time periods: January 2021 before the introduction of the vaccine; in June 2021 when the vaccine was given to elderly individuals and in September  when at least 70% of the population had at least one vaccination. Overall, I found the study to be well written with the authors pointing out strengths and potential caveats of this epidemiological study. I have just a few comments, which are listed below.

Comment 1, line 117-118: in the sentence, “the following scores were given: disagree = 1 point, disagree = 2 points, undecided = 3 points, agree = 4 points, and strongly 
agree = 5 points. Should the first score be “strongly disagree= 1 point.” 

Comment 2, line 127: The sentence with “(5) “ COVID-19 are easily spread from person to person," should be, “ (5) “ COVID-19 is easily spread from person to person," 

Comment 3, Table 2: Under household income, the “<400 Yen and >400 Yen” should be,  “<4 million Yen and >4 million Yen.

Comment 4, line 233: In the sentence reading, “between hesitancy and vaccine confidence and literacy were” would read better with “between hesitancy, vaccine confidence and literacy were.” 

Comment 5, line 268: In the sentence, “. In a study that analyzed twitter about COVID-19,”  twitter should be capitalized as it is name of a company.

Author Response

Thank you very much for reviewing our paper. We also thank you for your useful suggestions for improvement. We have made the following corrections.

Comment 1,

line 117-118: in the sentence, “the following scores were given: disagree = 1 point, disagree = 2 points, undecided = 3 points, agree = 4 points, and strongly

agree = 5 points. Should the first score be “strongly disagree= 1 point.”

Response,

We have corrected the points you raised. Please see line 124 of the manuscript.(in highlight)

Comment 2,

line 127: The sentence with “(5) “ COVID-19 are easily spread from person to person," should be, “ (5) “ COVID-19 is easily spread from person to person,"

Response,

We have corrected the points you raised. Please see line 134 of the manuscript. (in highlight)

Comment 3,

Table 2: Under household income, the “<400 Yen and >400 Yen” should be,  “<4 million Yen and >4 million Yen.”

Response,

We have corrected the points you raised. Please see Table 2 of the manuscript.

Comment 4,

 line 233: In the sentence reading, “between hesitancy and vaccine confidence and literacy were” would read better with “between hesitancy, vaccine confidence and literacy were.”

Response,

We have corrected the points you raised. Please see line 241-242 of the manuscript.

Comment 5,

line 268: In the sentence, “. In a study that analyzed twitter about COVID-19,”  twitter should be capitalized as it is name of a company.

Response,

We have corrected the points you raised. Please see line 276 of the manuscript.

Reviewer 2 Report

Although the paper is well written and properly structured, this reviewer needs to make some minor observations.

1. The abstract should highlight the innovative aspect of the study compared to previous studies.
2. The literature is well structured and complete. Please add some updated literature. I offer a few references below and I think it will be useful to add them ;

       Khurram Shahzad, Taimoor Hassan Farooq, Buhari Doğan, Li Zhong Hu & Umer   Shahzad (2021) Does environmental quality and weather induce COVID-19: Case study of Istanbul, Turkey, Environmental Forensics, DOI: 10.1080/15275922.2021.1940380

3. The databases are informative and adequate
4. The sources of data need to be cited properly in the paper – every reader should understand how to access these.
5. The research limitations should be stated clearly.
6. Policy implications of the results are weak and should be better articulated.

Author Response

Thank you very much for reviewing our paper. We also thank you for your useful suggestions for improvement. We have made the following corrections.

Comment 1,

The abstract should highlight the innovative aspect of the study compared to previous studies.

Response,

Thank you for your instruction. We have corrected the abstract. So, we inserted "In any phases" before "lower vaccine confidence" (line 26). Then we modified the last sentence. Please see line 29-33. (in highlight)

(before)

These results indicate that hesitancy increases transiently during vaccination introduction phases. We suggest that careful risk communication, including negative information, is necessary.

(after)

The findings suggested that hesitancy increases transiently during vaccination introduction phases, and changes as the vaccination program progressed or waves of epidemic. Careful risk communication to increase vaccine confidence and literacy is essential to reduce vaccine hesitancy, especially in introduction phase.

Comment 2,

The literature is well structured and complete. Please add some updated literature. I offer a few references below and I think it will be useful to add them;

       Khurram Shahzad, Taimoor Hassan Farooq, Buhari Doğan, Li Zhong Hu & Umer   Shahzad (2021) Does environmental quality and weather induce COVID-19: Case study of Istanbul, Turkey, Environmental Forensics, DOI: 10.1080/15275922.2021.1940380

Response,

Thank you for your guidance. I have added the following text in the limitation. Please see line 358-363 and line 488-491.

Third, it has been reported that temperature or air quality have a significant impact on COVID-19 mortality [42]; however, the influence of temperature and/or air quality on vaccine hesitancy and COVID-19 perception was not examined in the present study. Although regional differences in temperature exist in Japan, the temperature in June and September is similar within each region [43]. Thus, influence of temperature on present results may be minimal.

Comment 3,

The databases are informative and adequate

Response,

Thank you very much.

Comment 4,

The sources of data need to be cited properly in the paper – every reader should understand how to access these.

Response,

Thank you very much for your instruction.

We reviewed access to citations.

Comment 5,

The research limitations should be stated clearly.

Response,

Thank you very much for your instruction. We revised the limitation section shorter and clearer (line 357-363). (in highlight)

Comment 6,

Policy implications of the results are weak and should be better articulated.

Response,

We appreciate this constructive comment. We revised the conclusion section below (Line 368-372). (in highlight)

This study suggested that hesitancy increases transiently during vaccination intro-duction phases, and changes as the vaccination program progressed or waves of epidemic. In any phase, lower vaccine confidence and literacy showed a higher AOR of vaccine hesitancy. Careful risk communication to increase vaccine confidence and literacy is essential to reduce vaccine hesitancy, especially in introduction phase.

Reviewer 3 Report

In this manuscript, Tokiya et al. explored the vaccine hesitancy in Japan before vaccine approval, during the start of vaccination, and after implementation of vaccination. They have shown some interesting findings on how the perception of people changes during an ongoing pandemic and with different phases of vaccine introduction/launch. The introduction provides sufficient background information and rationale of the study, methods are sufficient, results and discussions are explained very well. Overall a very nicely written manuscript. I have few minor suggestions/concerns to improve the manuscript:

  1. In line 47, provide citations where it states 'according to previous studies'.
  2. It is not clear whether the participants were same at all 3 phases of the study. It would have been very interesting if the sample cohorts were same. 
  3. Regarding the 2.2.1 'definition of COVID-19 vaccine hesitancy', as per supplementary figure, those who were not eligible for vaccine were excluded from analysis. It is mentioned in methods but is unclear. Revise it to make clear that hesitancy has taken care of the fact that 'it is not because they did not have vaccine available for their age group'. 
  4. Line 117: looks like 1 point is for strongly disagree.
  5. Lines 133-141: these questions do not appear complete or are confusing the way they are presented now. Please revise to make clearer. 

Author Response

Thank you very much for reviewing our paper. We also thank you for your useful suggestions for improvement. We have made the following corrections.

Comment 1,

In line 47, provide citations where it states 'according to previous studies'.

Response,

We have corrected the points you raised. Please see line 50 of the manuscript. (in highlight)

Comment 2,

It is not clear whether the participants were same at all 3 phases of the study. It would have been very interesting if the sample cohorts were same.

Response,

Thank you for your instruction. We have corrected the following sentence in conjunction with reviewer 2's comments. Please see in line 357-358. (in highlight)

(before) Second, we analyzed the data in a cross-sectional manner, so we cannot be mentioned for causality.

(after) Second, the subjects of the three surveys are not the same, so it was not possible to examine changes over time.

Comment 3,

Regarding the 2.2.1 'definition of COVID-19 vaccine hesitancy', as per supplementary figure, those who were not eligible for vaccine were excluded from analysis. It is mentioned in methods but is unclear. Revise it to make clear that hesitancy has taken care of the fact that 'it is not because they did not have vaccine available for their age group'.

Response,

Thank you for your instruction. We have inserted the following minutes after “have no appointment date". Please see line 105-106.

It is not because they did not have vaccine available for their age group.

Comment 4,

Line 117: looks like 1 point is for strongly disagree.

Response,

We have corrected the points you raised. Please see line 124 of the manuscript.

Comment 5,

Lines 133-141: these questions do not appear complete or are confusing the way they are presented now. Please revise to make clearer.

Response,

Thank you for your instruction. We have removed the words in “()”, please see line 140-146.
